# Nano-Dimensional Carbon Nanosphere Supported Non-Precious Metal Oxide Composite: A Cathode Material for Sea Water Reduction

**DOI:** 10.3390/nano12234348

**Published:** 2022-12-06

**Authors:** Jayasmita Jana, Tran Van Phuc, Jin Suk Chung, Won Mook Choi, Seung Hyun Hur

**Affiliations:** School of Chemical Engineering, University of Ulsan, Daehak-ro 93, Nam-gu, Ulsan 44610, Republic of Korea

**Keywords:** amorphous carbon, oxide composite, electroactive surface area, seawater, electrocatalysis

## Abstract

Generation of hydrogen fuel at cathode during the electrolysis of seawater can be economically beneficial considering the vast availability of the electrolyte although it faces sluggishness caused by the anode reactions. In this regard a carbon nanosphere-protected CuO/Co_3_O_4_ (CCuU) composite was synthesized through heat treatment and was used as the cathode material for electrocatalytic seawater splitting. CCuU showed a significantly low overpotential of 73 mV@10 mA cm^−2^, Tafel slope of 58 mV dec^−1^ and relatively constant activity and morphology over a long time electrocatalytic study. A synergy within metal oxide centers was observed that boosted the proton-electron transfer at the active site. Moreover, the presence of carbon support increased the electroactive surface area and stability of the composite. The activity of the CCuU was studied for HER in KOH and alkaline NaCl solution to understand the activity. This work will pave the way for designing mesoporous non-precious electrocatalysts towards seawater electrocatalysis.

## 1. Introduction

Hydrogen, with high energy density [1], is an economical alternative fuel source for conventional and fossil fuel. Hence, the demand for facile and sustainable hydrogen generation is increasing rapidly. Currently, the steam methane reforming process and the coal gasification process for hydrogen generation are widely used for industrial applications. On the other hand, the associated hazards, including the release of toxic gases and waste materials, have advocated the need for a green chemical process [2]. In this respect, water electrolysis has become an important procedure for the sustainable evolution of hydrogen gas [3,4]. Electrocatalytic water splitting involves two half-cell reactions, the hydrogen evolution reaction (HER) and oxygen evolution reaction (OER). The large portions of HER research rely on fresh water at different pH. However, practically, the supply of this large amount of fresh water is troublesome in the era of water crises. In this regard, seawater, being an almost unlimited water source (~90%), could find its importance in HER research. However, the implementation of seawater as a source brings a few challenges in terms of the anodic reaction because of different ions (almost 50 % chloride, 48% sodium, and 2% other ions) [1,3,5]. During seawater electrocatalysis, in the anode, the OER faces challenges from the simultaneous generation of chlorine gas [6]. It has been suggested that the chlorine evolution reaction (CER) can be suppressed by the OER in a higher pH medium [7]. Despite the higher probability of the OER at a higher pH range, ${\mathrm{ClO}}^{-}$ formation remains a side reaction (${\mathrm{Cl}}^{-}+2{\mathrm{OH}}^{-}={\mathrm{ClO}}^{-}+H_{2}0+2e^{-}$) at the higher overpotential region (~480 mV) [8]. In addition to side reactions, the electrocatalysis of seawater suffers from electrode corrosion and poisoning of catalysts due to the precipitation of insoluble metal salts on the electrode surface [6]. These hazards limited the use of seawater for fuel cell applications. In this regard, scientists have taken some measures, such as ion-exchange membranes and reverse osmosis membranes, to control the movement of chloride ions towards the electrodes, thereby preventing the generation of toxic chlorine gas [9]. Nevertheless, these measures have their limitations. At the laboratory level, seawater is treated with a strong alkali to suppress Cl-related reactions at the electrode [10]. Initially, Pt was appointed as an efficient electrocatalyst for the HER, but its applications are limited by its low availability, low stability in alkaline media, and high cost [11].

Considering factors such as environmental effects and production cost, the design of non-precious metal (e.g., chalcogenides [1], nitride [4], phosphate [12], and hydroxide [3])-based electrocatalysts have been widely explored for seawater splitting by exploiting their active sites, near zero Gibbs free energy, effective electrolyte stability and thickness dependent behaviors. For example, Liu et al. reported that a CoP/Co_2_P heterostructured material acted as an efficient HER catalyst with an overpotential of 454 mV at 10 mA cm^−2^ in seawater media [13]. They formed a Mg/seawater battery with a porous CoP/Co_2_P hetero-structure cathode with a maximum power density of 6.28 mW cm^−2^ at a current density of 30 mA cm^−2^. Chen et al. fabricated a three-dimensional MoS_2_ quantum dot aerogel for the seawater HER. [11]. They observed that the aerogel remained efficient, even after 150 cycles of scanning, whereas commercial Pt/C showed lower activity after 50 cycles. Jin et al. synthesized two-dimensional Mo_5_N_6_ nanosheets and used the abundant metal-nitrogen electroactive sites on the material to achieve an effective HER in seawater [14]. Liu et al. synthesized a multi-phase hetero-structured CoNiP/Co*_x_*P for the alkaline seawater HER [15]. They reported an overpotential of 290 mV for 10 mA cm^−2^ current density during the HER using natural seawater. Yuan et al. synthesized bifunctional NiMo film for alkaline seawater (1M KOH + 0.5M NaCl) with an average cell voltage of ~1.563 V to reach a current density of 10 mA cm^−2^ with high durability [16]. Sarno et al. prepared a trimetallic NiRuIr-graphene nanocomposite electrocatalyst [17]. They found that the catalyst had a Tafel slope of 48 mV dec^−1^ and stability for almost 200 h without significant loss of efficiency. Huang et al. [18] designed a Fe-doped MoS_2_ nanosheet supported by three-dimensional carbon fibers to explore the HER in buffered seawater. The catalyst required overpotentials of 119 and 300 mV to reach 10 and 250 mA cm^–2^ current densities, respectively.

The method of engineering surface defects via heterojunction or lattice mismatch and alloying metal structures through the incorporation of heteroatoms has been used by scientists to improve the catalytic activity [19]. Moreover, in recent years, the use of carbon nanoparticle-based supports for metal catalysts has become a useful tool for improving the catalytic activity in terms of improved surface area, active sites, stability, and facile electronic interactions [20]. Jana et al. synthesized a carbon dot (CD)-associated Ni/Co hydroxide composite to achieve an efficient HER in both acidic and alkaline media [21]. The CDs provided facile electronic transfer efficiency and stability during long-term scanning. Liu et al. designed a N/S-doped carbon dot-supported Rh material for HER from seawater [22]. The presence of doped CDs promoted electrons at the Rh-CD surface and boosted the catalytic activity. The catalytic activity was comparable to the commercial Pt/C catalyst.

In this work, we have studied the effect of a carbon nanosupport on the metal composite for alkaline HER in seawater. For this purpose, a carbon nanoparticle-supported bimetallic oxide composite material was synthesized using a hydrothermal synthetic process followed by high-temperature calcination. Compared to common carbon dot supports, the as-synthesized amorphous carbon nanospheres could provide sufficient stability and synergism towards a bi-metal oxide composite (CuO/Co_3_O_4_). The presence of a carbon support on the metal oxide composite increased the electroactive surface area and catalytic activity despite the unaltered physical surface area. The catalytic activity of the as-synthesized material was checked using it as a cathode material during alkaline seawater splitting. To the best of our knowledge, this is the first time an amorphous carbon nanoparticle support has been used for the preparation of a non-precious metal cathode material for seawater electrocatalysis. The activity was tested using an alkaline NaCl electrolyte before using actual seawater (collected from the local sea). The activity was evaluated by measuring the overpotential for the reaction over a current density of 10 mA cm^−2^. Low overpotentials (73 mV and 115 mV for alkaline seawater and alkaline NaCl solution, respectively) were observed, which are one of the lowest values reported among overpotentials for seawater splitting (Appendix A). In addition, a chronoamperometric study showed that the stability of an as-synthesized catalyst was higher than that of the commercial Pt/C catalyst over 10 h (Appendix A). Thus, we have synthesized a carbon nanoparticle supported non-precious CuO/Co_3_O_4_ composite that can significantly contribute to the development of alternative fuel research through electrocatalytic seawater HER.

## 2. Experimental Section

### 2.1. Synthesis of Carbon Nanospheres (UCSs)

The nitrogen doped carbon nanospheres (UCSs) were synthesized using urea and triammonium citrate (TAC) as precursor compounds. In a typical hydrothermal synthetic route, 10 mL of 4.16 M aqueous solution of urea and 40 mL of aqueous solution of 1 M TAC were mixed and made a homogeneous solution through constant stirring. This solution was then transferred into a 100 mL Teflon-lined stainless-steel autoclave. The autoclave was then placed in a hot air oven for 8 h at 180 °C. After completion of the reaction, a brown colored solution was obtained. The solution was dialysed for 12 h (cutoff 1000 K MWCO). The as-obtained solution, termed UCS, was stored at room temperature for characterization and further applications. The UCS particles showed blue emission under UV-light. 

### 2.2. Synthesis of Carbon Nanosphere Functionalized Bimetallic Composite

The bimetallic composite has been synthesized using copper(II) chloride dihydrate (CuCl_2_·2H_2_O), cobalt(II) chloride hexahydrate (CoCl_2_·6H_2_O), urea, and as-synthesized UCSs. Typically, 5 mL of 0.88 M aqueous solution of CuCl_2_·2H_2_O, 5 mL of 0.63 M aqueous solution of CoCl_2_·6H_2_O, 1 mL of 1 M KOH solution, and 1 mL of 4.16 M aqueous solution of urea were mixed with continuous stirring. The mixture was transferred into a 50 mL Teflon-lined stainless-steel autoclave and kept in a hot air oven for 180 °C for 8 h. After completion of the reaction, the product was cooled at room temperature. The product was then centrifuged and washed several times to obtain the solid product. To this product, 5 mL of dialyzed UCS solution was added and stirred for 2 h. This mixture was then dried and calcined at 600 °C for 2 h. The as-obtained product was termed CCuU and stored at room temperature for characterization and electrochemical application. The C, C, u and U stands for Co, Cu, urea and UCS precursors. 

The effect of individual constituents was checked by the electrocatalytic behavior of the composite. For this purpose, the particular precursor material was omitted during above mentioned reaction steps. The compounds were termed CuuU, CouU, CCu, and CCU which were studied to understand the role of Co(II), Cu(II), UCSs (U) and urea (u). 

### 2.3. Study of Electrocatalytic Behavior

All electrochemical measurements were carried out using a three-electrode system consisting of a working electrode (WE), reference electrode (RE), and counter electrode (CE). A Hg/HgO electrode (at 1 M KOH solution) was used as RE, a graphite rod electrode was used as CE and a nickel foam (NF) with the surface area of 1.0 cm^−2^ was used as WE. The NFs were cleaned by sonication in HCl followed by water for 5 min each. The catalysts were dispersed in a nafion (5 % *v*/*v* solution in iso-propyl alcohol) solution at a concentration of 1 mg mL^−1^ through sonication for 30 min. The catalyst ink was drop-casted on NF and dried overnight with the optimized catalyst loading of 4 mg cm^−1^. The electrocatalytic activities were measured in terms of linear sweep voltammetry (LSV) scan within a range from 0 V to −0.6 V with respect to RHE at a rate of 5 mV s^−1^ in N_2_ saturated electrolyte (1 M KOH, alkaline NaCl, and alkaline sea water). The electrochemical impedance spectroscopy (EIS) studies were carried out at the frequency range of 100 kHz and 1 Hz at −0.62 V. The reference electrode potentials were converted in the reversible hydrogen electrode (RHE) scale by $E_{RHE}=E_{Hg/HgO}+0.921$. Here, $E_{RHE}$ is the converted potential. $E_{Hg/HgO}$ is the measured potential at RE. The Hg/HgO electrode was calibrated in 1M KOH to obtain the standard potential value of 0.921 V (Appendix A). The potential conversion depends on the pH of the medium. The overpotentials (η) were calculated with respect to the RHE. η can be related to the current density, J and Tafel slope, *b*, as η=b Log J+a. The electrochemically active surface area (ECSA) of the catalysts were measured by calculating electrochemical double-layer capacitance, $C_{dl}$, by $\mathrm{ECSA}=\frac{C_{dl}}{C_{s}}$. The cyclic voltammetry (CV) curves were recorded in potential range with a non-Faradaic current at different scan rates from 20 to 200 mV s^−1^. The slope of the relative current density vs. scan rate provided the value of $C_{dl}$. Here, the value of ‘*C_s_*’, specific capacitance of the catalyst per unit area under the same condition was taken to be 1.7 mF cm^−2^ as described by Panda et al. [23].

## 3. Results and Discussion

### 3.1. Characterization of the Composites and the Effect of UCSs

The 3D pricked bimetallic nanostructure was prepared using a urea templated method, as shown in Figure 1. First, the hydrothermal treatment of an alkaline solution of copper(II) chloride dihydrate and cobalt(II) chloride hexahydrate generated a needle-like morphology (Appendix A). The product was then reacted with pre-synthesized carbon nanospheres (UCSs), followed by high-temperature calcination. The product was termed CCuU. High temperatures caused the release of gases generated by the dissociation of urea, resulting in perforation of the material surface. The surface structure was confirmed by scanning electron microscopy (SEM), and transmission electron microscopy (TEM) showed perforation on the surface (Figure 2A,B). The SEM image revealed a sheet structure with a torn and punctured surface of the composite. The high resolution-TEM (HR-TEM) showed the presence of crystalline planes corresponding to planes of CuO [(111) [24,25]] and Co_3_O_4_ [(220) and (311)] [26,27] (Figure 2C). The lattice mismatch caused surface defects in the composite. The selected area electron diffraction (SAED) also indicated the presence of CuO (111), Co_3_O_4_ (220) and (400) crystal planes [28] (Figure 2D). Furthermore, the crystal planes of CCuU were studied by X-ray diffraction (XRD) over a range of 10° to 90° (Figure 3A). The pattern showed characteristic (−111), (200), (−112), (202), (−113), (−311), and (310) peaks for the monoclinic phase of CuO (Reference code: 01-080-1916) and (220), (311), (331), (422), (511) and (440) planes of cubic phase of Co_3_O_4_ (Reference code: 00-042-1467). Furthermore, the chemical states were analyzed by X-ray photoelectron spectroscopy (XPS), as shown in Appendix A. The broad range spectra revealed the presence of Co2p and Cu2p, O1s, and C1s. The atomic percentages were obtained from the XPS quantification (Appendix A). High resolution analysis of Co2p showed the presence of Co2p_3/2_ and Co2p_1/2_ of Co_3_O_4_ at 782.1 and 797.7 eV, respectively, with satellite peaks at 788.01 and 803.48 eV (Figure 3B) [29,30,31]. The spit-orbit energy separation of ~15 eV and the satellite peaks indicated a spinel structure of Co_3_O_4_ [31,32]. For Cu2p, two peaks at 955.8 and 935.8 eV corresponded to Cu2p_3/2_ and Cu2_p1/2_ of CuO [33], respectively. The satellite peak at 943.5 eV indicated Cu was present in +2 state (Figure 3C) [29]. Narrow range analysis of C1s revealed the presence of C=C, C=O, C–O, and O=C–O at 285.00, 285.37, 290.17, and 292.2 eV, respectively (Figure 3D) [34,35].

Electron microscopic studies of CCuU could not locate the precise position and corresponding lattice planes of carbon on the CCuU surface. The tiny size of the UCS particles compared to the micro-dimensional metal composite was not evident in the FESEM and TEM images. A thorough study of the structural properties of UCS particles (showed green emission at 520 nm when excited at 420 nm, Appendix A) revealed an average diameter of ~20–25 nm (Appendix A). The absence of definite lattice fringes and crystal planes (Appendix A) indicated that the UCS particles were amorphous. Elemental analysis of C1s during XPS revealed the presence of C–C, C=C, C=O, and O=C–O peaks (Appendix A) [35]. The XRD pattern (Appendix A) showed a broad peak around 20–26° 2θ (Reference code: 00-012-0212) for carbon; the broad peak indicated their amorphous nature [36]. Upon excitation at 532 nm, two distinct peaks at 1258 cm^−1^ and 1588 cm^−1^ were observed for UCS particles which might correspond to the D- and G- bands, respectively in Raman spectra. The shifted position of the D-bands indicated the increased size of UCS particles [37] and an oligomer structure [38]. The peak intensity decreased, and the peak position shifted after a reaction with the bimetallic motif (Appendix A). The Raman spectra indicated the attachment of carbon nanoparticles on the composite surface. Although the microscopic studies could not identify the precise position and crystal planes of carbon on the composite surface, the electrochemical activity and electrochemical surface area (ECSA) increased several-folds (detailed discussion follows).

The importance of UCSs on the structure and activity of CCuU was examined further by considering the bimetallic products prepared through similar reaction conditions except for the individual absence of urea and UCSs. The products are denoted as CCU and CCu, respectively. SEM revealed perforation on the CCu surface, whereas CCU exhibited an asymmetric morphology (Appendix A). Hence, the absence of urea did not produce sufficient roughness on the CCU surface, unlike CCu and CCuU. The XRD pattern indicated that CCu possessed crystal planes of CuO and Co_3_O_4_, whereas CCU possessed planes of CuO and CoO (Reference code: 00-042-1300), as shown in Appendix A. It can be concluded that UCS’s particles caused the oxidation of CoO to Co_3_O_4_ since the absence of UCS resulted in CoO only. The XPS of CCu (Appendix A) revealed the presence of Co2p, Cu2p, C1s, and O1s peaks in the broad spectrum. Further analysis of Co2p showed peaks for Co2p_3/2_ and Co2p_1/2_ of Co^2+^ at 782.7 and 798.2 eV, respectively. The spin orbit splitting of 16.1 eV between Co 2p_3/2_ and Co2p_1/2_ indicated +2 state of Co as CoO [39]. The Cu2p peaks were split into two main peaks of Cu2p_3/2_ and Cu2p_1/2_ at 936.3 and 956.3 eV, respectively. Similar peaks were observed for CCU at a broad range analysis (Appendix A). Splitting of the Cu2p peaks revealed the presence of two peaks for Cu2p_3/2_ and Cu2p_1/2_ at 935.4 and 955.6 eV, respectively. The nature of the metal oxides was further supported by the Raman spectral data (Appendix A). The F_2g_, E_g_ and A_1g_ bands of Co_3_O_4_ [40] were present in CCuU and CCu, while CCU contained peaks of CoO at 498, 543, and 698 cm^−1^ [41]; however, the CuO peaks at 296, 343, and 625 cm^−1^ [42] were present in these three materials. The results indicated that the presence of urea caused a difference in the reduction process and oxidation state of Co in the composite. During the reaction, urea decomposes to NH_3_ and CO_2_, which could work as a template and make the particles undergo Ostwald ripening for the formation of a hierarchical morphology (Appendix A) [43]. On the other hand, further addition of carbon nanospheres at higher temperature resulted in a carbon coating over the perforated composite structure. The oxidation states of elemental Co were modified in the presence of urea. Furthermore, the surface area of the composite increased when urea was used as a precursor. The Brunauer–Emmett–Teller (BET) surface areas (Appendix A), measured through nitrogen adsorption and desorption, were 27.22, 27.13, and 9.72 m^2^ g^−1^ with pore volumes of 0.362, 0.342, and 0.0706 cc g^−1^ for CCuU, CCu, and CCU, respectively (Appendix A). On the other hand, the ECSA was higher in the presence of UCSs (Appendix A). Carbon nanoparticles, being good electron transporters, could expand the contact area with the analyte, which in turn might increase the ECSA [44]. 

### 3.2. HER from Alkaline NaCl Solution

The electrocatalytic activity of these catalysts towards the hydrogen evolution reaction (HER) was studied using alkaline NaCl solutions (1 M NaCl solution in 1 M KOH medium) and alkaline seawater (1 M KOH) solution. The working electrode of the three-electrode system was a nickel foam coated with the catalyst materials with the catalyst loading of 4 mg cm^−1^. The catalytic activity was monitored in terms of the overpotentials at a certain current density by linear sweep voltammetry (LSV) at a scan rate of 5 mV s^−1^ over a potential range of 0 to −0.6 V vs. RHE. As evident from Figure 4 and Table 1, CCuU showed the lowest overpotential of 115 mV to obtain a current density of 10 mA cm^−2^, whereas commercial Pt/C and other catalysts required higher potential to reach that current density. The activity and mechanistic path can be understood from the Tafel slope. The corresponding Tafel slopes were calculated using extrapolation of the linear relationship between η and LogJ. Figure 4B shows that CCuU had a Tafel slope of 82 mV dec^−1^ (higher than CCu and Pt/C). The progress of the reaction could be understood from the Tafel slope values as they relate the potential required to increase the current density magnitude by ten-fold [45]. Since the Tafel slope value higher than 40 mV dec^−1^ can be a result from contribution of a higher coverage region it might be concluded that the either the Volmer step or Heyrovsky step at a higher coverage region act as the rate determining step [46].
(1)$$M+H_{2}O+e^{-}\to M-H+OH^{-}$$
(2)$$M-H+H_{2}O+e^{-}\to H_{2}+{\mathrm{OH}}^{-}+M$$
where M−H represents a hydrogen atom adsorbed on a catalyst surface. The rate-limiting step of alkaline NaCl water HER on CCuU might be related to the initial water dissociation and associated M−H adsorption [47]. The higher catalytic activity is often corroborated by the smaller charge transfer resistance [48]. Therefore, the electrochemical impedance (EIS) studies were performed to calculate the charge transfer resistance ($R_{ct}$). The corresponding Nyquist plot over a frequency range of 100 kHz to 1 Hz and amplitude of 7 mV (Figure 4C) indicated the smallest diameter of the semicircle for CCuU, with $R_{ct}$ = 1.0 Ω.

### 3.3. HER from Alkaline Seawater

Although the activities were significant for the seawater-like model, it is often difficult to achieve the HER on an actual seawater system due to stability and corrosion-related limitations [49]. Hence, the catalytic activities of the as-synthesized composites were tested on an actual seawater system under alkaline conditions, according to the above-mentioned experimental conditions. The seawater was treated with KOH solution, and the supernatant was used for the experiment. Figure 5 and Table 2 present the catalytic activities. As observed from the figure, the overpotential of 73 mV was required to attain 10 mA cm^−2^ current density by CCuU. This $\eta_{10}$ value is smallest along as-synthesized catalysts and much smaller than the commercial Pt/C ($\eta_{10}$ = 219 mV) under experimental conditions. However, it might be noticed that the $\eta_{100}$ was higher for CCU by 10 mV than CCuU. The activity was further understood from Tafel slopes. The Tafel slope for CCuU was 58 mV dec^−1^ whereas that for the Pt/C was 108 mV dec^−1^. The Tafel slope for CCU was 73 mV dec^−1^. This value of Tafel slope indicated that the Heyrovsky reaction might be the rate determination step with [46,50],

These values indicated a comparative activity of CCuU and CCU although Pt/C possessed lower activity compared to the as-synthesized composites. The EIS studies showed smaller $R_{ct}$ values for CCuU of 0.7 Ω compared to that of CCU (1.0 Ω) (Figure 5C). The results confirmed its highest activity of CCuU towards alkaline seawater through the fastest electron transfer. The catalytic activity of CCuU dominated over CCu, CCU, CuuU, and CouU, indicating that a proper composition of precursor materials results in a stable and electroactive material. It can further be noticed that the activity of catalysts was better for natural seawater than the seawater model. This might be due to the fact that in the seawater model or actual seawater electrolyte, there is more than one type of cation, and the electrolyte induces natural corrosion, which could be conquered by the CCuU through the carbon coating on the surface. The turnover frequency values (TOF) values were calculated at 200 mV to understand the efficiencies of conversion of substrates into H_2_. As shown in Table 3 and Appendix A, the CCuU (0.43 s^−1^) and CCU (0.5 s^−1^) have almost comparable activity explaining their similar rate of progress and direct influence of UCS particles. When compared with previous reports, it was found that the $\eta_{10}$ value is smaller than most of the previously reported materials. The high activity has been observed to be associated with significant stability (Appendix A). The above-mentioned results suggest that the as-synthesized catalyst CCuU exceed the activity of the commercial Pt/C. Furthermore, while constructing a two-electrode cell using IrO_2_ as the anode material, the cell potentials were calculated to be 1.593 V for CCuU and 1.683 V for Pt/C @ 10 mA cm^−2^ (Appendix A). These results indicate that CCuU could be successfully commercialized for hydrogen generation from alkaline seawater.

It was observed that CCU possessed a lower BET surface area than CCu, the electrocatalytic activity of CCU was higher than CCu in both cases and the η value was quite comparable with CcuU for alkaline seawater HER. This can be understood further by calculating the ECSA values [47]. The ECSA was calculated by measuring the double-layer capacitance, $C_{dl}$, at the potential window of 0.1 to 0.4 V using CV at different scan rates (Table 3) [51]. As shown in Appendix A, the porous CCuU showed the highest $C_{dl}$ values followed by CCU and CCu, indicating a larger number of exposed active sites and larger specific surface area resulting in improved HER activity [49]. The results suggested that the presence of the carbon nanoparticle support increased the active catalytic sites by increasing the electroactive surface area [20]. The improved HER for the actual seawater sample rather than the salt-treated freshwater might be due to the improved conductivity in the presence of spare metal ions, such as Na^+^, Mg^2+^, Ca^2+^, and K^+^ [52], than KOH and alkaline NaCl solutions [48]. As in the present case, the presence of Na^+^, Zn^2+^, and K^+^ ions on the CCuU surface was found from the XPS analysis (Appendix A). Furthermore, it has been previously reported that in many cases depending on the catalyst surface, NaCl concentration and amount of sacrificial agents might cause large differences in hydrogen production efficiency in natural seawater and synthetic seawater models [52].

### 3.4. Stability of CCuU

In addition to the activity, stability is also an important factor that decides the significance of catalysts during the electrolysis of seawater [49]. Figure 6 shows that during the chronopotentiometry study at a current of 130 mA over 10 h, the CCuU remained almost unchanged with a slight voltage drop of ~15 mV. The SEM image showed a perforated surface as before the HER. The XRD showed the retention of almost all peaks with slight deviations in peak intensities, which might be due to interactions with reaction intermediates. The high structural stability can be attributed to the presence of a conductive carbon layer on the composite surface which further offers plenty of active sites [21]. The effect of the protective carbon layer can further be understood by studying the fluorescence of carbon nanoparticles in the presence and absence of chloride ions (Appendix A). It was observed that even after attachment with chloride ions (as evident from FTIR studies) the fluorescence of carbon nanoparticles did not change due to the retention of fundamental electronic properties. The post-HER characterization did not indicate a significant change of the morphology and oxidation state, the voltage loss can be understood by the formation of a white precipitate with the progress of the reaction after 10 h, resulting in a variation of the current density over time. This precipitate was removed by adding a few drops of 0.5 M HCl solution. The removal of the precipitate is important to prevent the blocking of the active sites during long-term stability test [53].

### 3.5. Effects of Cl^−^ and Activity of CCuU

The effect of Cl- was understood by studying the catalytic activity in 1 M KOH which showed that the electrocatalytic activities of the materials differ in the absence of Cl^−^ or seawater in the alkaline medium. CCuU showed a higher overpotential of 128 mV, whereas commercial Pt/C showed a lower overpotential of 199 mV than that in alkaline seawater and alkaline NaCl solution. EIS revealed small semicircle curves for the catalysts (Figure 7, Table 4). This difference in activity in the alkaline medium only can be understood by considering the Cl^−^ ions present in the medium. The calculated Tafel slope value of CCuU (87 mV dec^−1^) indicated that the process would have a rate determining step of Volmer or Heyrovsky following a Volmer–Heyrovsky mechanism [47]. There might be factors, such as the co-adsorption of coexisting cations in seawater on the catalyst surface or in the WE itself [54,55] that can contribute to the rate determination. The HER in the alkaline medium is strongly affected by the H_2_O (H_2_O-M) and hydroxyl (OH-M) on the catalyst surface [15]. The presence of Cl^−^ in the medium affected the progress of the HER because Cl^−^ is a harder base than OH^−^ [3]. Moreover, the medium pH affected the catalytic activity transport of the ions [56]. On the other hand, based on the above-mentioned results and comparison of previous reports (Appendix A), the as-synthesized perforated CCuU nanocomposites would work as an efficient alternative to alkaline seawater hydrogen evolution.

## 4. Conclusions

In summary, a carbon nanosphere-supported CuO/Co_3_O_4_ bimetallic composite (CCuU) was successfully synthesized via heat treatment for electrolysis reaction. Detailed characterization showed that the presence of urea increased the surface roughness and carbon nanoparticles increased the electroactive surface area. The surface defects and the oxide induced ion transportation boosted the catalytic activity. The CCuU, when deposited on NF, exhibited significant catalytic activity and stability as a cathode material for HER in both alkaline saline water and alkaline seawater with much low overpotential (115 mV and 73 mV @ 10 mA cm^−2^) compared to most of the previously reported materials. The post-HER characterization of CCuU showed an almost unmodified physical state of the catalyst material. The stability may arise from the synergism within the metal oxides and the carbon coating. This study, describing the effects of amorphous carbon nanoparticles on the electrocatalytic activity of metal composites towards seawater splitting, would further contribute to research on seawater-based fuel cells as an alternative energy source.

## Figures and Tables

**Figure 1 nanomaterials-12-04348-f001:**
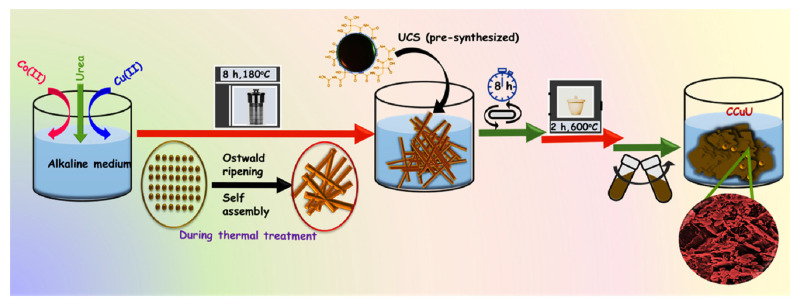
Schematic diagram of the synthesis of the CCuU composite. Other composites were synthesized through similar experimental process except the condition of precursor concentrations. The synthetic procedure has been provided in the supporting information.

**Figure 2 nanomaterials-12-04348-f002:**
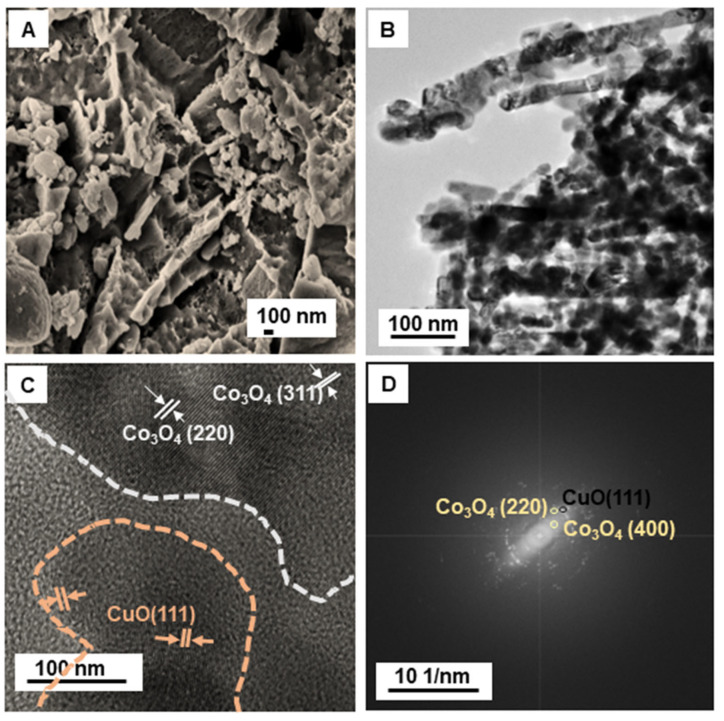
(**A**) SEM, (**B**) TEM, (**C**) HRTEM, and (**D**) FFT images of CCuU.

**Figure 3 nanomaterials-12-04348-f003:**
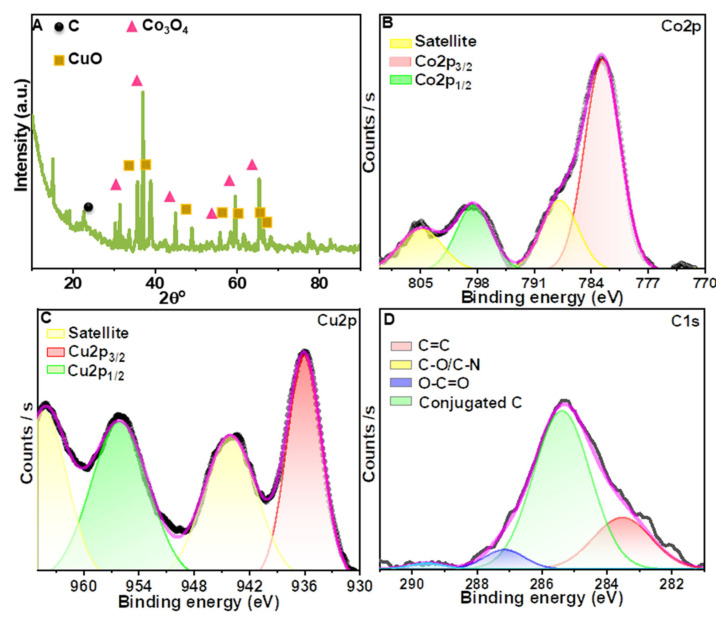
(**A**) The observed XRD pattern of CCuU. Narrow range XPS spectra of (**B**) Co2p, (**C**) Cu2p, and (**D**) C1s.

**Figure 4 nanomaterials-12-04348-f004:**
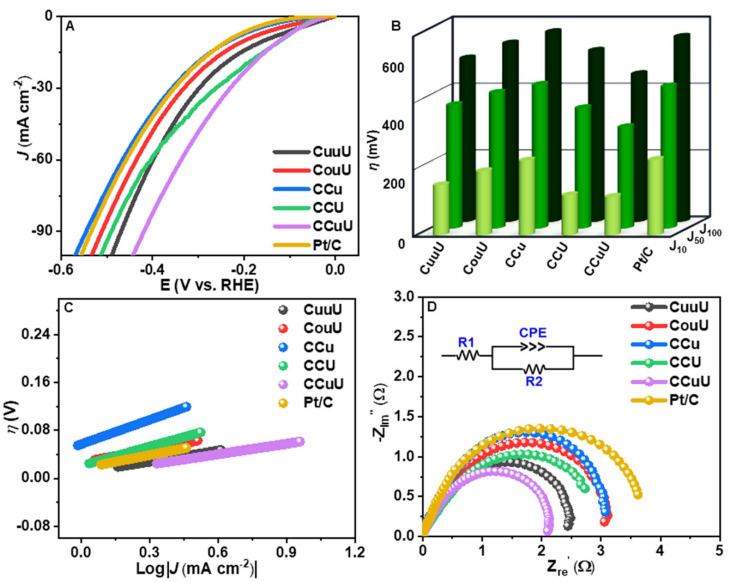
(**A**) LSV plots, (**B**) Comparison of overpotentials at different current densities, (**C**) Tafel plot, and (**D**) EIS diagram (inset: EEC) of catalysts in alkaline NaCl solution (1 M KOH + 1 M NaCl).

**Figure 5 nanomaterials-12-04348-f005:**
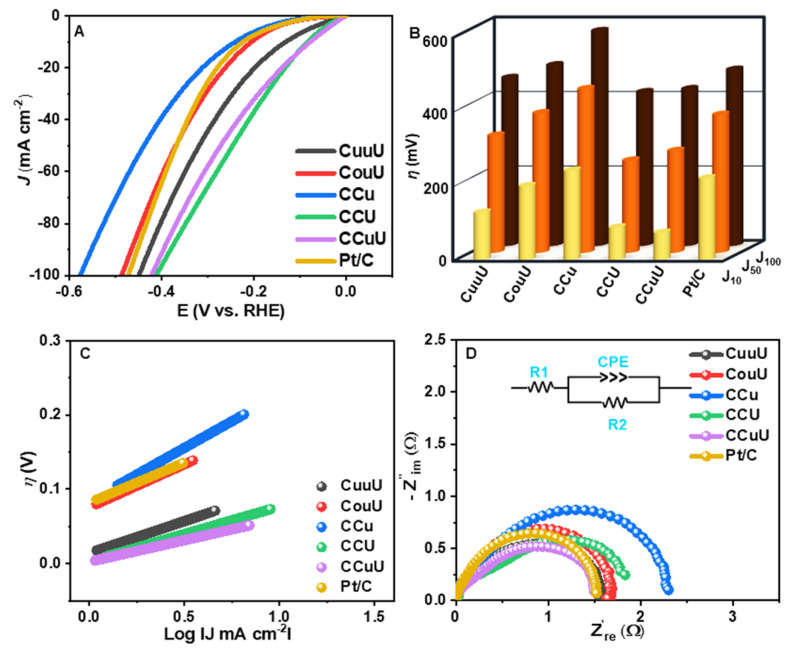
(**A**) LSV plots, (**B**) Comparison of overpotentials at different current densities, (**C**) Tafel plot, and (**D**) EIS diagram (inset: EEC) of catalysts in alkaline seawater (treated with 1 M KOH).

**Figure 6 nanomaterials-12-04348-f006:**
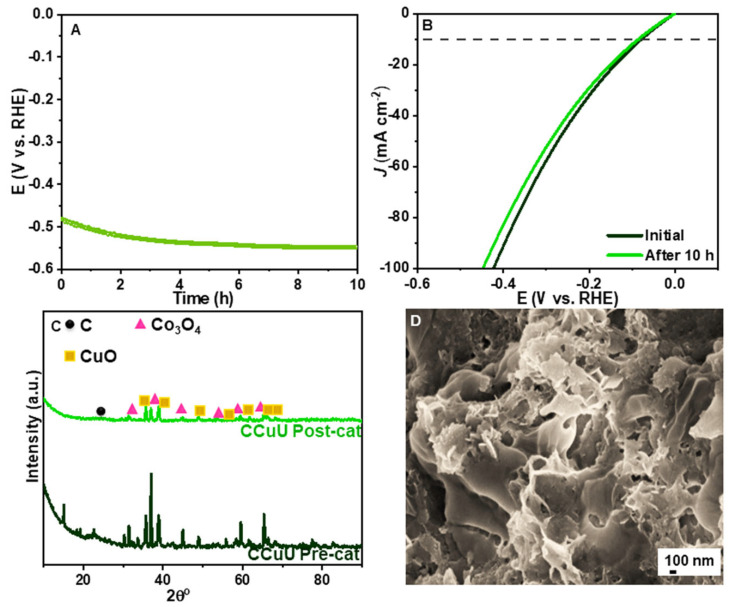
(**A**) Chronopotentiometry test of CCuU over 10 h (**B**) LSV curves before and after scanning. Post-HER (**C**) XRD pattern and (**D**) FESEM image of CCuU.

**Figure 7 nanomaterials-12-04348-f007:**
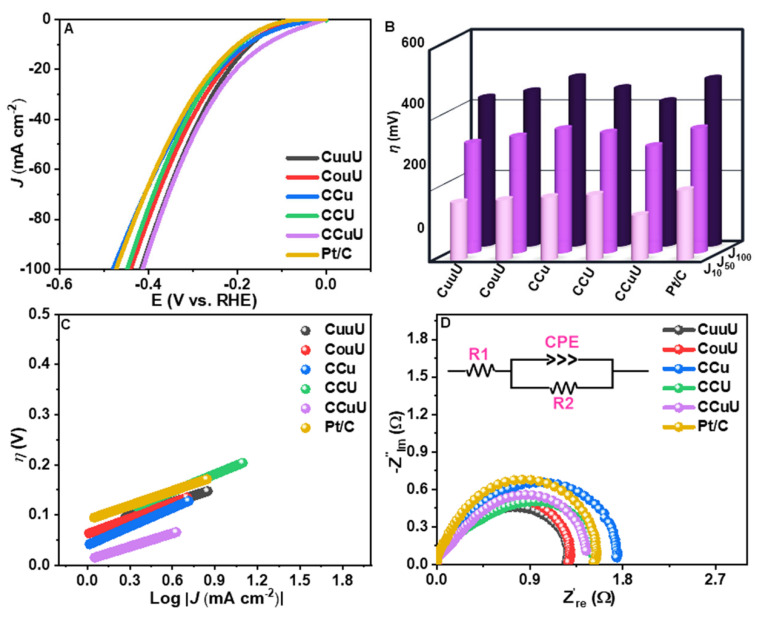
(**A**) LSV plots, (**B**) Comparison of overpotentials at different current densities, (**C**) Tafel plot, and (**D**) EIS diagram (inset: EEC) of catalysts in 1 M KOH.

**Table 1 nanomaterials-12-04348-t001:** Account of the catalytic activities of the materials in the alkaline NaCl solution.

**Catalysts**	η (mV) @10 mA cm^−2^	Tafel Slope mV dec^−1^	$R_{ct}\;\Omega$	$J_{0}\;\mathrm{mA}\;{\mathrm{cm}}^{-2}$
Pt/C	226	60	1.70	0.76
CuuU	150	70	1.20	1.51
CouU	192	135	1.48	0.59
CCu	224	105	1.47	0.46
CCU	120	64	1.43	0.77
CCuU	115	82	1.00	1.20

**Table 2 nanomaterials-12-04348-t002:** Account of the catalytic activities of the catalysts in alkaline seawater.

**Catalysts**	η (mV) @10 mA cm^−2^	Tafel Slope mV dec^−1^	$R_{ct}\;\Omega$	$J_{0}\;\mathrm{mA}\;{\mathrm{cm}}^{-2}$
Pt/C	219	108	0.8	0.679
CuuU	127	85	0.8	0.213
CouU	198	115	0.9	0.265
CCu	240	125	1.2	0.915
CCU	87	73	1.0	0.962
CCuU	73	58	0.7	0.173

**Table 3 nanomaterials-12-04348-t003:** Account of TOF alkaline seawater and electroactive sites of the as synthesized materials.

**Catalyst**	TOF **s^−1^**	$C_{dl}=\frac{d\left(\Delta J\right)}{2dV}$	$\mathrm{ECSA}\;{\mathrm{cm}}^{-2}$	$R_{f}=\frac{S_{\mathrm{ECSA}}}{S_{\mathrm{geo}}}$
CuuU	0.34	0.0199	11.7	11.7
CouU	0.19	0.0186	10.9	10.9
Ccu	0.09	0.0173	10.17	10.17
CCU	0.5	0.0270	15.8	15.8
CcuU	0.43	0.0302	17.7	17.7

**Table 4 nanomaterials-12-04348-t004:** Account of catalytic activities of the catalysts in 1 M KOH.

**Catalysts**	η (mV) @10 mA cm^−2^	Tafel Slope mV dec^−1^	$R_{ct}\;\Omega$	$J_{0}\;\mathrm{mA}\;{\mathrm{cm}}^{-2}$
Pt/C	199	96	0.64	0.131
CuuU	164	90	0.65	0.202
CouU	171	103	0.82	0.316
CCu	179	123	0.70	0.559
CCU	186	115	0.64	0.294
CCuU	128	87	0.70	0.961

## Data Availability

The data presented in this study are available on request from the corresponding author.

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
