# Peer review of "Nano-Dimensional Carbon Nanosphere Supported Non-Precious Metal Oxide Composite: A Cathode Material for Sea Water Reduction"

_nanomaterials, 2022, doi:10.3390/nano12234348_

Round 1

Reviewer 1 Report

This manuscript deals with the electrocatalytic activity for HER in both alkaline saline water and alkaline seawater of a carbon nanosupported bimetallic composite. The research is framed in terms of the development of sustainable hydrogen generation. The non-precious character of the obtained Cu/Co metal oxide composite, and also the study of the electrolysis of an abundant raw material such as seawater, make this investigation attractive. The research was conducted appropriately, the authors provide a detailed structural analysis and discussion of the characterization of the obtained material and the electrocatalytic studies. The results are presented in clear form and supported by the experiments. In my opinion, the manuscript deserves publication in Nanomaterials.

I only have some minor comments regarding the manuscript:

1)      In the HER experiments alkaline media is used to avoid the formation of chlorine, did the authors check the pH during the experiments? Are they constant or does it vary?

2)      Please check the incorrect use of 0 (zero) instead of O (letter) in the formulas: H2O, OH-, etc.

3)      The quality of the figures in PDF version created for revision is not good enough. It seems that the figures are out of focus.

Reviewer 2 Report

The manuscript entitled “Zero-Dimensional Carbon Nanosphere Supported Non-Precious Metal Oxide Composite: A Cathode Material for Sea Water Reduction”, reports an interesting research on the synthesis and characterization of carbon nanosphere functionalized bimetallic composite, to be used as cathode material for sea water reduction.

The manuscript is complete and clear both in the introduction and in the description of the experimental phases. The synthesized composites have been extensively characterized. The data are critically reported. The subject matter is topical and the results obtained are particularly interesting.

  I have no particular suggestions to send to the authors.

In light of the above, I believe that the manuscript can already be considered for publication

Reviewer 3 Report

The Authors report on carbon nanosphere-protected CuO/Co3O4 composite as cathode material to advance seawater electrolysis at low overpotential of 73 mV and Tafel slope 13 of 58 mV/dec. They compare the performance with several other composites - including the standard Pt/C - in alkaline saline water, alkaline seawater and alkaline water. The conclusions are reasonable and the achievements merit publication in Nanomaterials. I have only a few - mostly technical - points to clarify as listed below.

Title: Why 'Zero-dimensional'? Please delete this word from the title, because it is nor explained neither justified in the manuscript.

Fig. 1, Fig. 3, Fig. 4 and Fig. 5: the resolution in the proof is poor, please make sure that it will look fine in the final version.

In Fig. 3B the baseline was cut off, please change the y-axis scale for better visibilty.

Fig. 4C: The Tafel slopes was given in a different j regime for CCuU. Why? 

In the EIS plots the fitted curves are not visible. This should be fixed.

Fig. S12: in a 2-electrode system, the referencing is not relevant.

In Table S2 some overpotential values are given as negative numbers. I guess these are the potential values, please clarify.

line 183: 'Elemental analysis' is rather High-resolution, or Element specific 

line 185: ' spit-orbital separation' is spin-orbital

line 265: 'by 10 units' sounds more correct as 'by an order of magnitude'

The Authors describe seawater electrolysis using real seawater. Was the trace transition metal content checked? If not, did they check the elemental composition of the used electrode surface after electrolysis? Trace metal deposition on carbon materials may also boost performance.

Why did the Authors choose -0.62 V for EIS? Why not 10 mA/cm2 was the fix point, or a smaller, fixed overpotential value?

Reviewer 4 Report

The article presents interesting data on the electrocatalytic properties of carbon-mineral composites containing copper and cobalt oxides. The most important result presented in the article is the production of a number of cathode materials with low overvoltage that are sufficiently stable under alkaline sea water conditions.

The main remarks are related to the presentation of the results on the synthesis and characterization of the samples. This part of the article needs to be revised.

1. There is no information about the stage of sample synthesis in an autoclave. It is necessary to evaluate the pressure in the autoclave or give its characteristics (volume and filling with reagents) in more detail.

2. There is no information about the medium in which the samples were calcined at 600°C. When calcined in air, one can expect significant burnout in samples of amorphous carbon.

3. The abbreviations used in the article to designate various samples are unsuccessful. In particular, from the list of CuuU, CouU, CCu, and CCU, it is difficult to understand which composition each of them corresponds to. It is necessary to give a decoding for each such abbreviation, or provide a table with a decoding of all abbreviations used in the article.

4. There are no carbon nanospheres in the images shown in Figure 2 and Figure S6. This fact needs to be discussed. In this regard, the term “carbon nanosphere-supported CuO/Co3O4 bimetallic composite” used in this work cannot be considered correct.

5. It would make sense to carry out quantitative estimates for the phase composition of the studied samples (Figure S5 (A), Figure 6 (C)) and discuss the reasons for its change after the reaction (Figure 6 (C)).

Reviewer 5 Report

The authors report on design of a cathode composite material for electrochemical H2 generation from sea water. The results are very interesting and presented well. However, there are some points that need to be further elaborated before publishing the manuscript. 

XPS quantitative data on the Ni foam after ink preparation is needed to confirm the loading of Co and Ni.

It is not clear whether the authors performed any optimization for laoding of the Co and Ni containing materials.

What is the composition of the commercial Pt/C electrode?

XPS after HER measurements in seawater is needed to confirm the author's claim on co-adsorption of coexisting cations in seawater.

quality of the images are very low and need to be improved.

Round 2

Reviewer 4 Report

Necessary clarifications are included in the manuscript. The article may be published in the submitted form.

Reviewer 5 Report

The authors replied to the comments adequately. Therefore, I suggest the publication of the current version of the manuscript.